# Chain Model for Carbon Nanotube Bundle under Plane Strain Conditions

**DOI:** 10.3390/ma12233951

**Published:** 2019-11-28

**Authors:** Elena A. Korznikova, Leysan Kh. Rysaeva, Alexander V. Savin, Elvira G. Soboleva, Evgenii G. Ekomasov, Marat A. Ilgamov, Sergey V. Dmitriev

**Affiliations:** 1Institute for Metals Superplasticity Problems, Russian Academy of Sciences, Khalturin St., 39, 450001 Ufa, Russia; elena.a.korznikova@gmail.com (E.A.K.); lesya813rys@gmail.com (L.K.R.); 2Institute of Physical Chemistry of RAS, Kosygin St., 4, 119991 Moscow, Russia; asavin00@gmail.com; 3Yurga Institute of Technology (Branch), National Research Tomsk Polytechnic University, 652050 Yurga, Russia; sobolevaeno@mail.ru; 4South Ural State University (National Research University), Lenin Ave., 76, 454080 Chelyabinsk, Russia; ekomasoveg@gmail.com; 5Institute of Mechanics, Ufa Federal Research Center, Russian Academy of Sciences, Oktyabrya Ave., 71, 450054 Ufa, Russia; ilgamov@anrb.ru; 6National Research Tomsk State University, Lenin Ave., 36, 634050 Tomsk, Russia

**Keywords:** carbon nanotube bundle, plane strain conditions, lateral compression, equilibrium structure, thermal stability, chain model, 61.48.De

## Abstract

Carbon nanotubes (CNTs) have record high tensile strength and Young’s modulus, which makes them ideal for making super strong yarns, ropes, fillers for composites, solid lubricants, etc. The mechanical properties of CNT bundles have been addressed in a number of experimental and theoretical studies. The development of efficient computational methods for solving this problem is an important step in the design of new CNT-based materials. In the present study, an atomistic chain model is proposed to analyze the mechanical response of CNT bundles under plane strain conditions. The model takes into account the tensile and bending rigidity of the CNT wall, as well as the van der Waals interactions between walls. Due to the discrete character of the model, it is able to describe large curvature of the CNT wall and the fracture of the walls at very high pressures, where both of these problems are difficult to address in frame of continuum mechanics models. As an example, equilibrium structures of CNT crystal under biaxial, strain controlled loading are obtained and their thermal stability is analyzed. The obtained results agree well with previously reported data. In addition, a new equilibrium structure with four SNTs in a translational cell is reported. The model offered here can be applied with great efficiency to the analysis of the mechanical properties of CNT bundles composed of single-walled or multi-walled CNTs under plane strain conditions due to considerable reduction in the number of degrees of freedom.

## 1. Introduction

There exist a huge number of carbon polymorphs, including a wide class of sp^2^ structures such as fullerenes, carbon nanotubes (CNT), and graphene. Due to the action of relatively weak van der Waals forces, a great variety of secondary structures can be formed, and some of them can have a long-range order, for example, fullerite crystal composed of fullerenes [1,2,3], graphite made of graphene layers [4,5], and crystals made of CNTs [6,7,8]. Such crystalline structures are of great interest since they have properties not exhibited by isolated structural elements [1,2,3,4,5,6,7,8]. Here, we focus on mechanical response of CNT bundles.

Various experimental techniques have been developed to produce CNT forests [9,10,11,12]. Mechanical applications of CNTs include ropes [7,13], fibers [14,15,16,17,18], polymer-matrix and metal-matrix composites [19,20,21], solid lubricants [21,22], etc. In all these applications, superior mechanical properties of CNTs such as tensile strength in the range from 11 to 63 GPa, tensile Young’s modulus of the order of 1.0 to 1.3 TPa, and high deformability up to ultimate fracture strain of about 10% are used [23,24,25,26]. In addition, they are lightweight, flexible, have high thermal and electrical conductivity, and these properties are useful in a number of applications [27,28,29,30].

Not only tension [7,13,14,15,16,17,18] and compression of vertically aligned CNT brushes and forests [31,32,33,34,35,36,37,38], but also lateral compression of isolated CNT or CNT bundles [39,40,41,42,43] is of interest, and the latter loading scheme has been studied less thoroughly than the former ones. Drawing, winding, micromechanical rolling, and shear pressing were used to produce horizontally aligned CNT bundles from vertically aligned CNT arrays [44,45,46,47]. Experimental and computational approaches used for evaluation of mechanical properties of CNTs have been outlined in the review [48]. Carbon nanotube bundles are linear elastic under hydrostatic pressure up to 1.5 GPa at room temperature; the volume compressibility, measured by in situ synchrotron x-ray diffraction, is 0.024 GPa^−1^; the deformation of the trigonal nanotube lattice under hydrostatic pressure is reversible up to 4 GPa [49]. Using X-ray diffraction and Raman scattering techniques, it has been shown that CNT bundles under non-hydrostatic pressure are not reversible for pressures beyond 5 GPa [50].

Indentation experiments are widely used to assess mechanical properties of vertically aligned CNT forests and brushes [32,33,34,35,36,37,38]. In particular, the Young’s modulus of 200 nm thick brush is about 17 GPa and the critical buckling stress can be estimated as 0.3 GPa at a load of 0.02 mN [32]. Carbon nanotube forests composed of CNTs of 2.2 mm height and 50 nm diameter have shown an elastic compressive modulus of 2.1 MPa as measured at initial loading condition and 20.8 MPa as measured after the plateau region [33].

Computational studies on the mechanical properties of nanomaterials become increasingly important because they speed up and reduce the cost of research and design. On the other hand, it is a tremendous challenge to predict nonlinear mechanical behaviors of nanomaterials by full atomic molecular dynamics method due to the huge computational cost, particularly for CNT bundles. Development of new computational approaches capable of modeling mechanical response at different scales is a very important task.

Mesoscopic modeling of phase transformations and mechanical deformation mechanism of CNT forest has been addressed in [51,52]. It has been shown that under compression along the tubes a low-density phase composed of vertically aligned CNT bundles transforms into a dense phase with horizontal alignment of CNTs. Carbon nanotubes subject to large deformations obtain different morphological patterns that can be simulated using a continuum shell model [53]. Transverse mechanical properties of CNTs have been studied in [8]. Nonlocal beam, plate, and shell theories employed in modeling of the mechanical properties of nanoscale structures are described in the review [54]. The applicability of the continuum beam model in the mechanics of CNT has been discussed in [55]. It has been shown that the rigidity of CNT crystal does not decrease with increasing tube diameter [6]. Isolated CNT of diameter above a threshold value can have two stable configurations—circular and collapsed [56,57,58]. In a recent experimental and molecular dynamics study, the irreversible transformation of triple-wall carbon nanotube bundles have been analyzed at pressure up 72 GPa and temperature up to 2400 K [39]. The irreversible transformation threshold pressure has been found to be in between 60 GPa and 72 GPa. Nonlinear coarse-grained stretching and bending potentials for CNTs have been developed to enable simulation of the mechanical behaviors and failure mechanism of the CNT bundles [59].

In spite of the fact that a number of computational methods have been developed for the analysis of mechanical properties of CNT forests, there is always a need to increase the efficiency and accuracy of simulation methods. Continuum mechanics is a powerful and effective tool to successfully describe macroscopic parameters of CNT bundles, but it has some limitations. For example, thermal fluctuations of CNTs and their fracture can be more adequately modelled in frame of atomistic models. On the other hand, full atomic models, as mentioned above, are very demanding on computational resources. One possible compromise is to use atomistic models for particular deformation modes, when the number of degrees of freedom can be substantially reduced.

In this study, in order to reduce the number of degrees of freedom, a full atomic model of CNT bundles under plane strain conditions is substituted by the chain model developed in the work [60] and successfully used to study structure and properties of secondary structures such as folds and scrolls of carbon nanoribbons [60,61,62,63,64] and dynamics of surface ripplocations on a graphite substrate [65]. For simplicity, here we will only consider the case of a bundle composed of single-walled CNTs of equal diameter, but the model can be applied to the cases of CNTs of different diameter, multi-walled CNTs, and include graphene scrolls and cylindrically crumpled graphene.

## 2. Materials and Methods

The computational model employed in this study is schematically shown in Figure 1. The nanotube bundle is aligned along the *z*-axis and CNTs of equal diameter create in cross-section a triangular lattice; they are numbered by the indices i=1,…,I and j=1,…,J (the case of I=J=2 is shown). Only zigzag CNTs are considered for simplicity. The carbon atoms move on the (x,y) plane and each atom represents a rigid row of atoms oriented normal to the (x,y) plane. Within each CNT, carbon atoms are numbered by the index n=1,…,N, anti-clockwise, starting from the atom with maximal *x*-coordinate. Thus, total number of atoms in the computational cell is I×J×N. Atomic positions are defined by the radius-vectors $r_{ijn}=\left(x_{ijn},y_{ijn}\right)$. Periodic boundary conditions are imposed.

Let us describe the model geometry. The interatomic distance in graphene is equal to ρ=1.418 Å. The distance between neighboring atomic rows oriented along the armchair direction in graphene is then $a=\rho \sqrt{3}/2=1.228$ Å, and this is the distance between atoms in the chain model (see Figure 1). Carbon nanotube diameter is D=a/sin(π/N). Let *d* be the shortest distance between CNT walls, then the distance between centers of neighboring CNTs is A=D+d. The sides of the computational cell in the form of parallelogram are I×A and J×A. In our simulations, we consider CNTs with N=30 having diameter D=11.75 Å and equilibrium value of d=3.088 Å, which can be compared to the interplanar distance of graphite equal to 3.3 Å.

Carbon nanotube bundle under uniform lateral compression can be efficiently described by the Hamiltonian of the chain model [60] (1)$$H=K+U_{B}+U_{A}+U_{\mathrm{VdW}},$$
which includes kinetic energy (2)$$K=\frac{M}{2}\sum_{i=1}^{I}\sum_{j=1}^{J}\sum_{n=1}^{N}\left({\dot{x}}_{ijn}^{2}+{\dot{y}}_{ijn}^{2}\right),$$
energy of valence bonds (3)$$U_{B}=\sum_{i=1}^{I}\sum_{j=1}^{J}\sum_{n=1}^{N}V(|r_{ijn+1}-r_{ijn}\left|\right),\;\mathrm{where}\;V\left(r\right)=\frac{k}{2}{\left(r-a\right)}^{2},$$
energy of valence angles (4)$$U_{A}=\sum_{i=1}^{I}\sum_{j=1}^{J}\sum_{n=1}^{N}P\left(\theta_{ijn}\right),\;\mathrm{where}\;P\left(\theta \right)=\epsilon \left[\cos\left(\theta \right)+1\right],$$
and energy of van der Waals interactions (5)$$U_{\mathrm{VdW}}=\sum_{i=1}^{I}\sum_{j=1}^{J}\sum_{n=1}^{N}\sum_{i^{\prime }=1}^{I}\sum_{j^{\prime }=1}^{J}\sum_{n^{\prime }=1}^{N}W(|r_{ijn}-r_{i^{\prime }j^{\prime }n^{\prime }}\left|),\;\mathrm{where}\;\right|n^{\prime }-n|>3\;\mathrm{when}\;i=i^{\prime },\;\;j=j^{\prime }.$$

In Equation (Equation 2) *M* is the carbon atom mass, which is 12 amu. In our simulations, time is measured in picoseconds, energy in eV, and distance in angstrom. In these units $M=12\times 1.0364\times 10^{-4}$. As it can be seen from Equation (Equation 3), harmonic potential with stiffness *k* is used to model deformation of the valence bonds. In order to reproduce the longitudinal stiffness of graphene sheet one should take k=405 N/m [60], which in the units adopted here gives k=25.279.

In Equation (Equation 4), cosine of the angle between two valence bonds, $r_{ijn}-r_{ijn-1}$ and $r_{ijn+1}-r_{ijn}$, is calculated as (6)$$\cos\left(\theta_{ijn}\right)=\frac{\left(r_{ijn}-r_{ijn-1},r_{ijn+1}-r_{ijn}\right)}{|r_{ijn}-r_{ijn-1}\left|\right|r_{ijn+1}-r_{ijn}|}.$$
Bending rigidity of graphene sheet is well reproduced with the value of the potential parameter ϵ=3.50 eV [60].

The van der Waals interactions in Equation (Equation 5) are given by the Lennard–Jones potential (5,11) [65] (7)$$W\left(r\right)=\frac{\varepsilon }{6}\left[5{\left(\frac{\sigma }{r}\right)}^{11}-11{\left(\frac{\sigma }{r}\right)}^{5}\right],$$
with the interaction energy ε=0.00166 eV and the equilibrium bond length σ=3.61 Å.

Further information on the chain model and on the procedure of fitting its parameters can be found in [60,65].

As it has been mentioned, a CNT of sufficiently large diameter can have either cylindrical or collapsed equilibrium configuration. In the present study we consider CNTs of relatively small diameter (N=30, D=11.75 Å) with only circular stable state when unloaded.

The aim of this study is to evaluate mechanical response of CNT bundle to lateral biaxial compression under plane strain condition with $\varepsilon_{xx}=\varepsilon_{yy}\leq 0$ and $\varepsilon_{xy}=0$. Firstly, equilibrium configurations are found at zero temperature and then, their stability at room temperature (T=300 K) is analyzed.

Perturbation–relaxation molecular dynamics simulations are done at zero temperature in order to find equilibrium structures at different values of applied strain and T=0 K. The simulation protocol is as follows. The compressive strain is applied by increments $\Delta \varepsilon_{xx}=\Delta \varepsilon_{yy}=-0.0025$ starting from zero strain. After each increment, the positions of atoms are perturbed by adding small random displacements to their *x*- and *y*-coordinates. The displacements are uniformly distributed in the range from $-10^{-6}$ to $10^{-6}$ Å. Then the equilibrium structure is obtained by minimizing potential energy of the system with the help of the gradient method. Energy minimization stops when the absolute value of the maximal force acting on atoms becomes smaller than $10^{-10}$ eV/Å.

Different computational cell sizes were considered. Calculations with 24×24=576 CNTs have revealed that structures with period doubling are formed as a result of instability at particular value of compressive strain. Such structures can be analyzed with smaller cell size and most of the results reported here are for the cell that includes 6×6=36 CNTs.

Classical molecular dynamics was used to assess stability of equilibrium structures with respect to thermal fluctuations at T=300 K. Temperature in our simulations is defined as (8)$$T=\frac{M}{2IJNk_{B}\left(t_{2}-t_{1}\right)}\int_{t_{1}}^{t_{2}}\sum_{i=1}^{I}\sum_{j=1}^{J}\sum_{n=1}^{N}\left({\dot{r}}_{ijn},{\dot{r}}_{ijn}\right)d\tau \;,$$
where $k_{B}=8.617\times 10^{-5}$ eV·K^−1^ is the Boltzmann constant and the averaging time is $t_{2}-t_{1}$ = 10 ps. For a given temperature *T*, the initial velocities of atoms are assigned according to the Maxwellian distribution. Random initial displacements of atoms are assigned in a way to increase the potential energy of the system by the amount equal to the kinetic energy.

Equations of atomic motion that stem from the Hamiltonian Equation (Equation 1) are integrated numerically with the help of the Stormer method of order six with the time step of 0.1 fs. The structure is considered to be stable if no structure transformations are observed within 100 ps. Structure transformations can be very well seen on the time dependencies of kinetic and potential energies during our simulations with NVE ensemble (constant number of particles, volume, and total energy). When structure changes, kinetic energy increases in expense of potential energy.

## 3. Results

In this section the equilibrium structures of CNT bundle under lateral compression are reported and their properties are analyzed. First, the potential energy and stress as functions of strain are given and then the change of CNT geometry with strain is presented. Finally, stability of equilibrium structures at T=300 K is analyzed.

### 3.1. Energy and Stress in the System

We start with the analysis of potential energy per atom calculated for equilibrium structures at different values of compressive strain. In Figure 2a, total potential energy per atom is shown as a function of strain, while in Figure 2b–d this energy is decomposed into three parts: the energy of van der Waals interactions, the energy of valence bonds, and the energy of valence angles, respectively. Total potential energy in the range of strain below 3.75% increases with strain quadratically but for larger strain a linear increase of energy with strain can be observed. These two regimes are separated in Figure 2 by the vertical dashed line. This qualitative change in the behavior of potential energy is due to structural changes observed in the system for strain exceeding the critical value of 3.75%. Below this critical strain, all CNTs in the system have the same cross-section in the form of six-fold flattened cylinders (see Figure 3a), while above the critical value the CNTs become elliptic. Two different structures with elliptic CNTs can form, Structure I with the translational cell doubled in one direction (see Figure 3b) and Structure II with the translational cell doubled in two directions (see Figure 4). In Figure 2 results for Structure I are shown by red circles and for Structure II by black triangles.

From Figure 2a–d it is clear that below the critical strain the potential energy of all three kinds increase with strain. At the critical value of strain just before the structure transformation one has $U_{\mathrm{VdW}}=0.0041$ eV, $U_{B}=0.0056$ eV, and $U_{A}=0.0029$ eV. The largest increase of energy is observed for the valence bonds because main mechanism of lattice deformation in this regime is contraction of valence bonds. The smallest contribution to the energy increase comes from the valence angles, since they do not change much during transformation of CNT cylindrical shape into six-fold flattened shape. This picture drastically changes for compressive strain larger than 3.75% when CNTs become elliptic. The deformation of structure in this regime is mainly due to change of valence angles and $U_{A}$ increases rapidly with strain while other two components of energy decrease with strain. The decrease of $U_{\mathrm{VdW}}$ with strain in this regime is explained by formation of new van der Waals bonds with increasing ellipticity of CNTs. At strain of 8.75%, $U_{A}$ is already one order of magnitude larger than two other components of energy. Note that both Structure I and Structure II have very close energies, and only for strain above 7% is $U_{\mathrm{VdW}}$ in Structure I slightly higher than in Structure II. Repetition of the simulations with different random atomic displacements has revealed that Structure I and Structure II are formed at the transition point with nearly equal probability, and this is because they have practically same energy.

Variation of stress components $\sigma_{xx}$, $\sigma_{yy}$, and $\sigma_{xy}$ with strain is shown in Figure 2e–g, respectively. For strain below the critical level, compressive stress increases so that $\sigma_{xx}=\sigma_{yy}$ and $\sigma_{xy}=0$. This is because the structure with six-fold symmetry is elastically isotropic (see Figure 3a). For strain above the critical level, deformation occurs at nearly constant pressure $p=-(\sigma_{xx}+\sigma_{yy})/2$. Both structures become anisotropic. In particular, Structure I is orthotropic with $|\sigma_{xx}\left|>\right|\sigma_{yy}|$ and $\sigma_{xy}=0$. On the other hand, Structure II has general anisotropy with $|\sigma_{xx}\left|<\right|\sigma_{yy}|$ and $\sigma_{xy}\neq 0$.

### 3.2. Geometry of CNTs

To better understand the relation between structure and macroscopic parameters of the system, let us quantify the geometry of CNTs in different structures. For each CNT we calculate its minimal and maximal diameters, $D_{\min}$ and $D_{\max}$, and the angle α between the *x*-axis and the maximal diameter, as shown in Figure 3b.

In Figure 5a,b the ratio $D_{\min}/D_{\max}$ is shown for Structures I and II, respectively. In Figure 5c,d the angle α is shown for Structures I and II, respectively.

Since the translation cell of Structure I includes two CNTs (numbered as 1 and 2 in Figure 3b), in Figure 5a,c, two different values of $D_{\min}/D_{\max}$ and α can be seen for strain above the critical value. The translation cell of Structure II includes four CNTs (numbered as 1 to 4 in Figure 4) and hence, in Figure 5b,d, four different values of $D_{\min}/D_{\max}$ and α can be seen for strain above the critical value. For strain below the critical value, all curves merge into one since translation cell includes single CNT, see Figure 3a.

From Figure 5a, one can see that in Structure I the ellipticity of SNTs gradually increases with increasing compressive strain. At the value of strain 8.75%, $D_{\min}/D_{\max}$ is smaller than 0.5. Recall that further increase of strain above 9% results in the formation of collapsed CNTs with non-convex cross-section, but we do not analyze such structures here. As Figure 5c suggests, the orientation of elliptic CNTs in Structure I is practically strain-independent. Carbon nanotubes 1 and 2 have orientation angles α=35 and 145 deg., with the difference equal to 110 deg.

Structure II demonstrates more complicated evolution with strain.

Two different regimes can be distinguished looking at Figure 5b. For strain between 4% and 6.5%, three CNTs (1, 2, and 4) in the translation cell have nearly the same ellipticity, while CNT 3 is less elliptic (see Figure 4a). For larger strain, CNTs 2 and 4 become more elliptic than CNTs 1 and 3 (see Figure 4b). Orientation angles of CNTs also change with strain (see Figure 5d). At 8.75% strain, CNT 4 is nearly aligned with the *y*-axis (α is close to 90 deg.), while CNT 2 with the *x*-axis (α is close to 180 deg.). Carbon nanotubes 1 and 3 have angles of about 50 and 130 deg., with a difference of about 80 deg.

### 3.3. Temperature Effect

The stability of all three types of equilibrium structures reported in Section 3.1 with respect to thermal oscillations is investigated here at temperature T=300 K.

It was found that the structure with all identical CNTs (see Figure 3a) is stable at room temperature up to compressive strain of about 4.0%, which slightly exceeds the stability range of this structure at 0 K (3.75% strain). At a strain of 4.0% and temperature 300 K, the cross-sectional shapes of CNTs fluctuate in time but on average all CNTs remain the same, preserving the high symmetry of the structure. The pressure-induced phase transition from high-symmetry structure to the structures with period doubling in one or two directions is the second-order phase transition. This is justified by the absence of the jumps in macroscopic properties at the transition point (3.75% compressive strain), see Figure 2. Low-symmetry Structures I and II under heating transform into high-symmetry structures, if the compressive strain is not too high.

Within the range of compressive strain from 4.1% to 6%, Structures I and II are stable at 300 K; they are preserved within the simulation time of 100 ps and no jumps of macroscopic parameters are observed.

Both Structures I and II become unstable at 300 K for compressive strain exceeding 6%. The instability of Structure I is illustrated in Figure 6 for compressive strain of 7%. In (a,b), one can see the time evolution of kinetic energy per atom and components of compressive stress, respectively. Until t=10 ps, kinetic energy oscillates near the value of 0.0259 eV, which corresponds to 300 K, but then it starts to increase in expense of the potential energy (total energy is conserved in the system). Pressure drops at the transition point from 175 to 140 MPa. Jumps in the macroscopic parameters of the system indicate that this phase transition is of the first order. In (c), a snapshot of the structure is presented at t=20 ps. One can see that the long-range crystal order is lost and an irregular structure that includes collapsed CNTs with non-convex cross-section is formed.

## 4. Discussion

The chain model introduced in the works [60,65] was developed here to enable simulation of the mechanical properties of CNT bundles under plane strain conditions. The model was applied to the analysis of structure transformations and mechanical properties of CNT crystal subjected to biaxial lateral compression. Carbon nanotube diameter is relatively small, so that the collapsed shape is unstable in the absence of external forces.

Three different crystalline structures stable at zero temperature have been found. For compressive strain $|\varepsilon_{xx}\left|=\right|\varepsilon_{yy}|$ < 3.75%, primitive translational cell of the crystal includes single CNT. In the range of compressive strain from 4% to 9%, two phases of nearly the same potential energy were found, one has two and the other one has four CNTs in a primitive translational cell. These structures are referred to as Structure I and II, respectively. Pressure-induced phase transition from the high-symmetry structure to the Structures I or II is of the second order because no jumps of macroscopic properties are seen in Figure 2 at the transition point at strain of 3.75%.

Thermal fluctuations increase the stability range of the high-symmetry structure with single CNT in the primitive translational cell. At 300 K, this structure is stable up to 4.0% compressive strain, while at 0 K the stability threshold is at 3.75% strain. The transformation of low-symmetry phase into high-symmetry phase under an increase in temperature is typical for the second-order phase transitions [66,67].

Thermal fluctuations reduce the stability range of Structures I and II. At zero temperature, they are stable from 3.75% to 8.75% of compressive strain, while at room temperature the stability range of compressive strain is from 4.1% to 6%. The transition above 6% strain is of the first order, it is accompanied by the jumps in macroscopic parameters (see Figure 6a,b), when crystalline structure with a long-range order transforms into an irregular structure (see Figure 6c).

As for the mechanical properties of CNT bundles under lateral compression, the transition to the structures with elliptic CNTs results in a considerable drop in rigidity of the bundle. Indeed, the deformation of the structure in the range from 3.75% to 9% compressive strain is at nearly constant pressure, see Figure 2e,f. Unloading of the system from any strain below 9% has shown that the structure is non-linear elastic with no hysteresis effect. Also note that the high-symmetry structure is elastically isotropic, while Structures I and II are anisotropic since for them $\sigma_{xx}\neq \sigma_{yy}$, see Figure 2e,f.

The results reported here are in agreement with the results of full atomic and continuum mechanics modelling [6,8,39,51,53,54,56,57,58], but they were obtained at a very low computational cost. For example, full atomic modelling with the same accuracy would require the use of the periodic boundary conditions in the *z* direction with at least one zigzag carbon chain within the translational cell. The calculation of interatomic forces between atoms within the considered cell and its translation images would require additional summation, which is absent in the chain model since it has been done in derivation of the effective potentials between the rigid atomic chains oriented along the *z* axis. For this reason, the chain model gives at least one order of magnitude acceleration of computations with the same accuracy, as compared to full atomic modelling.

Recall that the harmonic, unbreakable potential is used in this work to describe the valence interactions between carbon atoms, which is sufficient for modelling structure transformations at relatively small pressure considered here. In order to model irreversible deformation of CNT bundles under very high pressure, see, e.g., the work [39], the harmonic potential Equation (Equation 3) should be substituted with the breakable anharmonic potential, such as Morse potential [68].

In future works, it is important to study the effect of CNT diameter since new effects can be expected for larger diameter when collapsed isolated CNT is stable. Crystals composed of such CNTs can demonstrate irreversible plastic deformation with very peculiar mechanisms of plasticity. As for the applications, CNT bundles under lateral compression can show hysteresis effect, when it acts as an elastic damper [69], in a similar way to the compressed, vertically aligned CNT brushes [31].

The chain model proposed here can be readily adjusted to a number of newly found graphene-analogous 2D nanomaterials [70] by fitting model parameters to the results of first-principle or molecular dynamics simulations.

Consideration of bundles composed of different CNTs or multi-walled CNTs is straightforward. Substitution of the harmonic valence bond potential in Equation (Equation 3) with a suitable breakable anharmonic potential will enable the simulation of structure transformations in the CNT bundle under very high pressure.

## 5. Conclusions

We thus conclude that the chain model can be applied with a high numerical efficiency and sufficient accuracy to the analysis of structural and mechanical properties of CNT bundles under plane strain conditions.

The atomistic chain model proposed here, unlike continuum mechanics models, is able to describe high curvature of collapsed CNT wall and fracture of the walls under high pressure.

The chain model proposed here can be readily applied to the cases of CNTs of different diameter, collapsed CNTs, multi-walled CNTs, and even include graphene scrolls and cylindrically crumpled graphene.

## Figures and Tables

**Figure 1 materials-12-03951-f001:**
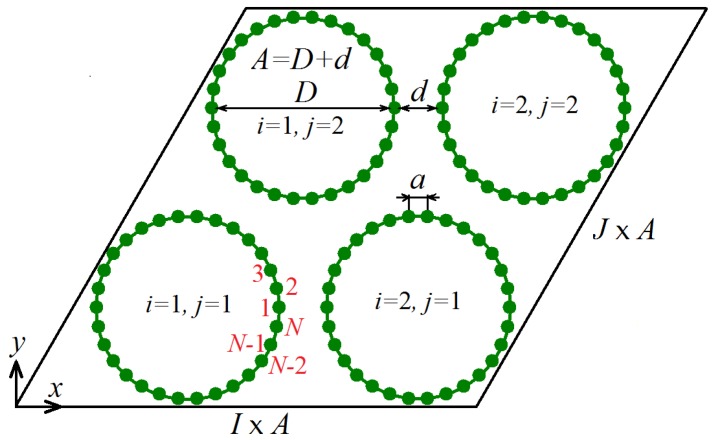
Schematic of the computational cell that includes I×J carbon nanotubes (CNTs) (I=J=2 in this case) numbered by the indices i=1,…,I and j=1,…,J. Carbon nanotubes in cross-section create a triangular lattice. Within each CNT, carbon atoms are numbered by the index n=1,…,N anti-clockwise, starting from the atom with maximal *x*-coordinate. Atoms move on the (x,y) plane. Each atom represents a row of atoms oriented normal to the (x,y) plane, which moves as a rigid body. The computational cell has the shape of a parallelogram with the sides I×A and J×A, where *A* is the distance between centers of neighboring CNTs. Periodic boundary conditions are used.

**Figure 2 materials-12-03951-f002:**
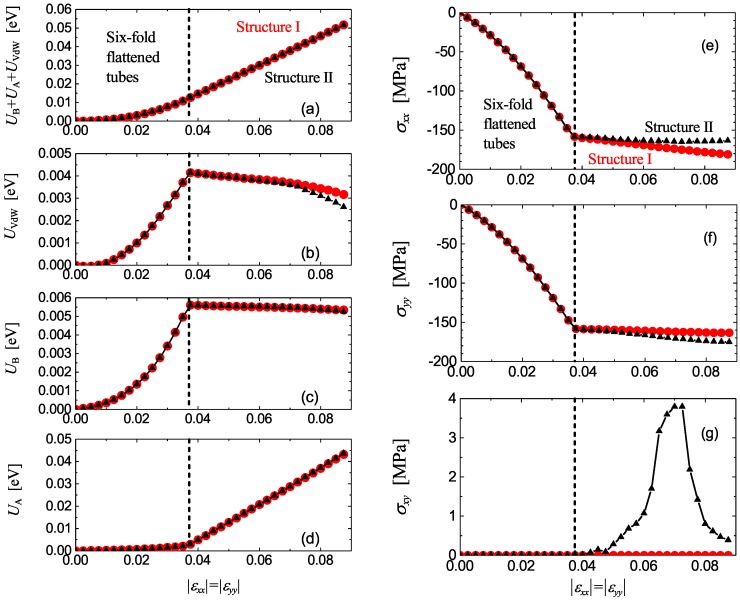
(**a**–**d**) Potential energy per atom and its parts as the functions of biaxial compressive strain. (**e**–**g**) Components of stress tensor as the functions of biaxial compressive strain. Results for Structure I [see Figure 3b] are shown by red circles and for Structure II (see Figure 4) by black triangles. These structures with elliptic nanotubes are stable for $0.0375<|\varepsilon_{xx}\left|=\right|\varepsilon_{yy}|<0.09$. For smaller values of compressive strain one has six-fold flattened nanotubes [see Figure 3a]. For compressive strain above 9% collapsed CNTs appear in the system (this regime is not studied here). The first critical value of strain is shown by the vertical dashed lines.

**Figure 3 materials-12-03951-f003:**
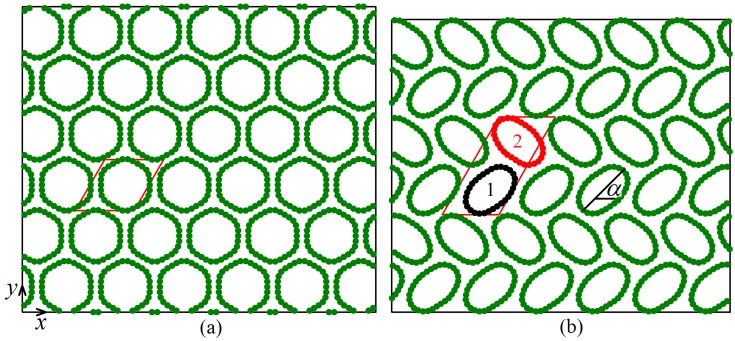
Equilibrium structures of CNT bundle observed at compressive strain of (**a**) 3.5% and (**b**) 5.5%. In (**a**) the displacements of atoms are multiplied by factor 4 to better reveal the six-fold flattened cylindrical shape of CNTs. In (**b**) the CNT cross-section is elliptic. Translation cells of the structures are shown by red lines. In (**a**) the cell includes single CNT, while in (**b**) period doubles in one direction. The latter structure is referred to as Structure I.

**Figure 4 materials-12-03951-f004:**
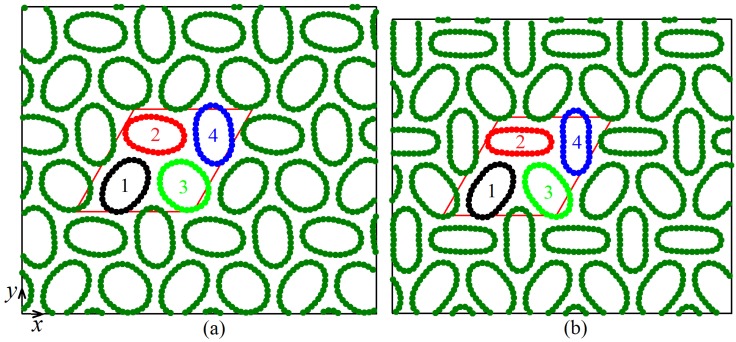
Equilibrium structures of CNT bundle observed at compressive strain of (**a**) 5.5% and (**b**) 8.5%. Translation cells of the structures are shown by red lines. Here, period doubles in two directions and this structure is called Structure II.

**Figure 5 materials-12-03951-f005:**
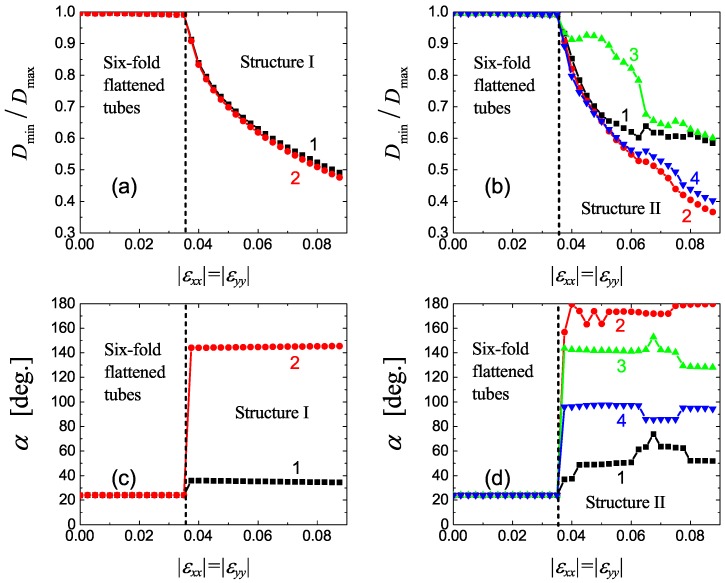
(**a**,**b**) The minimal to maximal diameter ratio for CNTs in Structures I and II, respectively, as functions of compressive strain. (**c**,**d**) Orientation angle of CNTs in Structures I and II, respectively, as functions of compressive strain. Numbers near the curves link them to the CNTs in translation cells of Structures I and II, as shown in Figure 3b and Figure 4. The critical value of strain is shown by the vertical dashed lines.

**Figure 6 materials-12-03951-f006:**
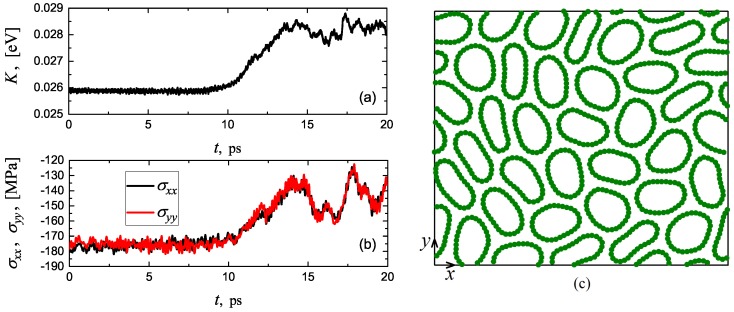
Instability of Structure I at compressive strain of 7% and temperature T=300 K. (**a**,**b**) Time evolution of kinetic energy per atom and components of compressive stress, respectively. Structure transformation begins at t≈10 ps, which results in the change of macroscopic parameters. (**c**) Snapshot of structure at t=20 ps. As a result of structure transformation, collapsed, non-convex CNTs appear in the system.

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
