# Peer review of "Chain Model for Carbon Nanotube Bundle under Plane Strain Conditions"

_materials, 2019, doi:10.3390/ma12233951_

Round 1

Reviewer 1 Report

-Conclusions should be separate future challenges and not contain references. They should reflect the main findings of this work

-The sentence "Not only tension but also lateral compression of CNT bundles is of interest, and the latter loading scheme has been studied not as thoroughly as the former one" should be detailed and cited properly.

-Abbreviations and acronyms should be used in full when they appear in the beginning of sentence

-Experimental work should be included in introduction, e.g. CNT forest mechanical properties assessment though nanoindentation

-"very low computational cost" should be evidenced and detailed as statement, comparing to e.g. a given case

Author Response

Response to Reviewers’ comments and description of the amendments applied to the manuscript

We would like to thank the Reviewer for prompt action and for the constructive criticism and useful suggestions.

First Reviewer wrote:

-Conclusions should be separate future challenges and not contain references. They should reflect the main findings of this work

Our response: We have rewritten Conclusions. The description of the future work was moved to the end of the Discussion. Citation of references was removed from the Conclusions.

First Reviewer wrote:

-The sentence "Not only tension but also lateral compression of CNT bundles is of interest, and the latter loading scheme has been studied not as thoroughly as the former one" should be detailed and cited properly.

Our response: Now this statement is supported by the references:

“Not only tension [7,13-18] and compression of vertically aligned CNT brushes and forests [31-38], but also lateral compression of isolated CNT or CNT bundles [39-43] is of interest, and the latter loading scheme has been studied not as thoroughly as the former ones.”

First Reviewer wrote:

-Abbreviations and acronyms should be used in full when they appear in the beginning of sentence

Our response: In eight occasions we have substituted CNT with “Carbon nanotube” in the beginning of sentence.

First Reviewer wrote:

-Experimental work should be included in introduction, e.g. CNT forest mechanical properties assessment though nanoindentation

Our response: In the Introduction we have added citations to the experimental works on mechanical properties of CNT forests and brushes:  

“Indentation experiments are widely used to assess mechanical properties of vertically aligned

CNT forests and brushes [32-38]. In particular, the Young’s modulus of 200 nm thick brush is about 17 GPa and the critical buckling stress can be estimated as 0.3 GPa at a load of 0.02 mN [32]. Carbon nanotube forest composed of CNTs of 2.2 mm height and 50 nm diameter have shown the elastic compressive modulus of 2.1 MPa as measured at initial loading condition and 20.8 MPa as measured after plateau region [33].”

First Reviewer wrote:

-"very low computational cost" should be evidenced and detailed as statement, comparing to e.g. a given case

Our response: The following arguments have been added after this statement:

“For example, full atomic modelling with the same accuracy would require the use of the periodic boundary conditions in $z$ direction with at least one zigzag carbon chain within the translational cell. Calculation of interatomic forces between atoms within the considered cell and its translation images would require additional summation which is absent in the chain model since it has been done in derivation of the effective potentials between the rigid atomic chains oriented along $z$ axis. For this reason, the chain model gives at least one order of magnitude acceleration of computations with the same accuracy, as compared to full atomic modelling.”

Reviewer 2 Report

The paper reports a computational study on carbon nanotube bundle under plane strain conditions. The topic is interesting for readers of Materials. The paper is well organized and the results are properly presented and discussed. The paper can be published after the following minor revisions

The novelty of the proposed model should be better highlighted in the Abstract as well as in the Conclusions. Is it possible to ipothesize a similar approach on nanoparticles with different morphology? Please discuss it in the Introduction.

Author Response

Response to Reviewers’ comments and description of the amendments applied to the manuscript

We would like to thank the Reviewer for prompt action and for the constructive criticism and useful suggestions.

Second Reviewer

The novelty of the proposed model should be better highlighted in the Abstract as well as in the Conclusions. Is it possible to ipothesize a similar approach on nanoparticles with different morphology? Please discuss it in the Introduction.

Our response: We agree. The following sentence was added to the Abstract in order to highlight the novelty of the model:

“Due to the discrete character of the model, it is able to describe large curvature of CNT wall and the fracture of the walls at very high pressures, both these problems are difficult to address in frame of continuum mechanics models.”

In Conclusions we state:

“The atomistic chain model proposed here, unlike continuum mechanics models, is able to describe high curvature of collapsed CNT wall and fracture of the walls under high pressure.”

In the Introductions (the last sentence) we describe other morphologies that can be readily addressed in frame of proposed chain model:

“For simplicity here we will only consider the case of the bundle composed of single-walled CNTs of equal diameter, but the model can be applied to the cases of CNTs of different diameter, multi-walled CNTs and include graphene scrolls and cylindrically crumpled graphene.”